# An Islanding Signal-Based Smooth Transition Control in AC/DC Hybrid Micro-Grids

**Zhenghong Chen [1,2], Tao Zheng [1,*] and Chang Liu [3]**

[1]   Shaanxi Key Laboratory of Smart Grid, Xi'an Jiaotong University, Xi'an 710049, China
[2]   NR Electric Co., Ltd., Nanjing 211102, China
[3]   State Grid Jibei Electric Power Company Limited Management Training Center, Beijing 102401, China
*   Correspondence: tzheng@mail.xjtu.edu.cn; Tel.: +86-029-82663692



**Featured Application: The hybrid micro grid structure, control strategy and smooth transition control proposed in this paper can be applied in small scale low-voltage hybrid micro-grids, such as an industrial plant with both AC and DC equipments.**

**Abstract:** Smooth transition is one of the most important issues of micro-grids. The resulting transition is much impacted by the state step of the regulator. To suppress this mutation, this paper proposes a smooth transition control based on an islanding signal, which updates the state of the regulators by detecting the change of islanding signal. Pre-synchronization control is applied during the transition from islanding mode to grid-connected mode. In comparison, the proposed approach is superior over direct transition control and state follower-based transition control, with both easier regulator parameter configuration and better performance during transition time.

**Keywords:** hybrid micro-grid; droop control; smooth transition; islanding signal

## 1. Introduction

The electricity demand has greatly increased with the rapid development of society and the economy. This exposes some problems of the fossil fuel-enabled power supply system—for example, the overuse of fossil and air pollution. Thus, it stimulates the utilization of renewable energies in large-scale centralized power plant and distributed power generators. Due to the intermittent feature of renewable energy, large-scale integration may bring instability of the grid. The micro-grid as a union of distributed power sources, energy storage systems, energy conversion devices, loads, supervisory systems, and protection has attracted more attention for its highly efficient energy utilization and flexible control [1–3].

The operation, control, and protection of alternating current (AC) micro-grids has been conducted by the researchers for a long time [4]. Recently, with the increase of direct current (DC) power equipment, such as mobile phones, computers, electric vehicles, frequency conversion devices, and so on, the concept of the DC micro-grid has been proposed to reduce redundant AC/DC converters [5,6]. To further improve the power transfer efficiency, an AC/DC hybrid micro-grid has been proposed [7]. A hybrid micro-grid has both an AC bus and a DC bus, and they are interconnected through an interlink converter (ILC). A hybrid micro-grid provides an integration of all kinds of distributed generators (DGs) and loads. The number of converters can be reduced, and thus the power quality is improved. As a result, it becomes a prospective structure of distributed energy integration.

A micro-grid has two operation modes: grid-connected mode and islanding mode [8–10]. In most cases, the micro-grid operates in grid-connected mode. When an external fault happens or the power quality cannot meet the requirements, the micro-grid disconnects from the utility grid by switching at

the point of common coupling (PCC), and enters islanding mode to provide continuous power supply for the local load, in order to have a reliable power supply. After the external fault is cleared, the micro-grid switches back to grid-connected mode [11,12].

The transition may occur from grid-connected mode to islanding mode, or inversely from islanding mode to grid-connected mode. During the transition from grid-connected mode to islanding mode, due to the different control strategy, a new power management will be achieved; thus, the transition impacts occur. During the transition from islanding mode to grid-connected mode, besides the power management problem, synchronization transition impacts occur as well. Impulse current affects current quality, and may cause damage to electrical equipment [13,14]. It has a bad effect on power grid stability, and in some extreme situations, it may even lead to splitting of the power grid. Therefore, it is important to provide a smooth transition control between different operational modes of the micro-grid.

In [15], transient impulses during the smooth transition between on-grid and off-grid status are investigated regarding a demonstration project of the micro-grid at Polytechnic of Bari, Italy. This micro-grid is integrated into a low voltage distribution network, including programmable or nonprogrammable generators, loads, and storage systems. To reduce the impacts of impulses, it employs a supervisory control and data acquisition system to optimize the inner micro-grid power production, in order to minimize the exchange power with utility grid [15,16]. Thus, a smooth transition may be achieved. However, most of the time, exchange power between micro-grid and utility grid is not easily controlled, as expected. In various power exchanges, a smooth transition is always required. In [17], voltage control in islanding mode and current control in grid-connected mode are performed, while an extra synchronous control strategy is used during the transition period to suppress transient bumps. This kind of control performs well with the small amount of power exchange between the micro-grid and utility grid. In [18], a diesel generator is used to compensate for power changes during transition period. In contrast to electronic converter-based generators, it takes time for rotary machines to output the required power. Therefore, the way to minimize the exchange power may take effect in a smooth transition. On the other hand, a control strategy may be modified to satisfy a smooth transition. In [19], an adaptive droop coefficient method is proposed to mitigate voltage deviation, thereby reducing the transient impact. In [20,21], the focus is on a smooth transition control method based on a regulation state follower when switching from grid-connected mode to islanding mode. There are two controllers for two different working modes. The proposed smooth control method makes use of the free controller to follow the state of the working one, ensuring that the output of two controllers stay equal. This does work in some situations. However, it takes a long time to realize the state following. In addition, the control parameters are hard to be set to fit for different working modes, or even worse, it happens that no parameters are available.

As a result, during the transition period, a smooth control that can both deal with the various exchanges of power between micro-grid and utility grid, and respond instantly with easy controller configuration, is expected. To this end, this paper proposes a smooth transition approach based on the islanding signal. It focuses on the regulator of the control system instead of power exchange of the whole system. In the proposed control strategy, the state of the controller is renewed if an islanding signal is detected during the transient period. In this way, the state step of the controller will be completely suppressed, which usually exists in the traditional state follower-based transition control. Furthermore, setting the control parameters becomes much easier than that in the state follower-based approach. Since the state step has been suppressed, it takes a short time to realize the smooth transition.

The rest of this paper is organized as follows. Section 2 builds a model of AC/DC hybrid micro-grid for a small plant. In Section 3, the transition control strategy is described in detail. Simulation results and impacts on the proposed approach are discussed in Section 4, and conclusions are drawn in Section 5.

## 2. Hybrid Micro-Grid Modeling

Regarding a real plant in a southern county of Jiangsu Province, China, an AC/DC hybrid micro-grid model has been built, as shown in Figure 1, where PV represents photovoltaic panel, DC represents the direct current, AC represents the alternating current, PCC represents the point of common coupling, and ILC represents the interlink converter.

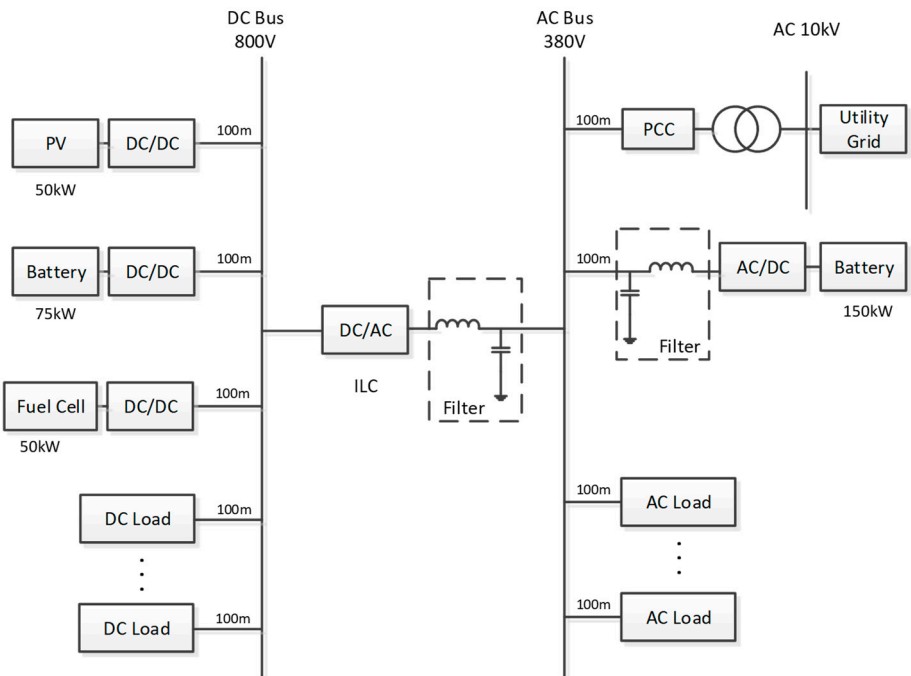

**Figure 1.** Hybrid micro-grid structure of a small plant.

In grid-connected mode, the utility grid provides the voltage and frequency references for AC sub-network. Renewable sources in the AC or DC sub-network work in MPPT(Maximum Power Point Tracking) mode to generate maximum power. The energy storage component absorbs power to charge, then shuts off when it is fully charged. The DC sub-network is connected to the AC bus through an ILC, and operates as an active load of AC sub-network. In a grid-connected operation, the AC sub-network, DC sub-network, and utility grid can realize the free flow of power in the entire network through an ILC and PCC. Power fluctuations caused by output changes and load changes can be balanced.

When fault happens in the utility grid, the PCC switch receives a signal and disconnects the micro-grid; then the hybrid micro-grid starts to work under a mixed islanding mode. DGs in an AC sub-network work under P-V, Q-f droop control to provide voltage and frequency reference for the AC system. DGs in a DC micro-grid work under P-V droop control to provide voltage reference for the DC system. The offset of AC and DC voltage represents the load condition in sub-networks. By measuring the offset of AC and DC bus voltage, and putting it into the control, free power exchange can be achieved between the AC and DC sub-network. In this way, a droop control in the entire network can be realized [22], as shown in Figure 2. Herein, ESS indicates energy storage system and DG indicates distributed generator.

The droop lines of AC and DC sources can be gathered in the same axis, as shown in Figure 3. The ILC serves as the bridge between the AC and DC sub-networks, and plays an important role in the power exchange on both sides. Expected load sharing in the entire micro-grid can be achieved by controlling AC unit voltage to be equal to DC unit voltage. Because the voltage value corresponds to the consumed power, the outer power loop can be modified to the voltage loop. The ILC control strategy is shown in Figure 4, where PWM means pulse-width modulation.

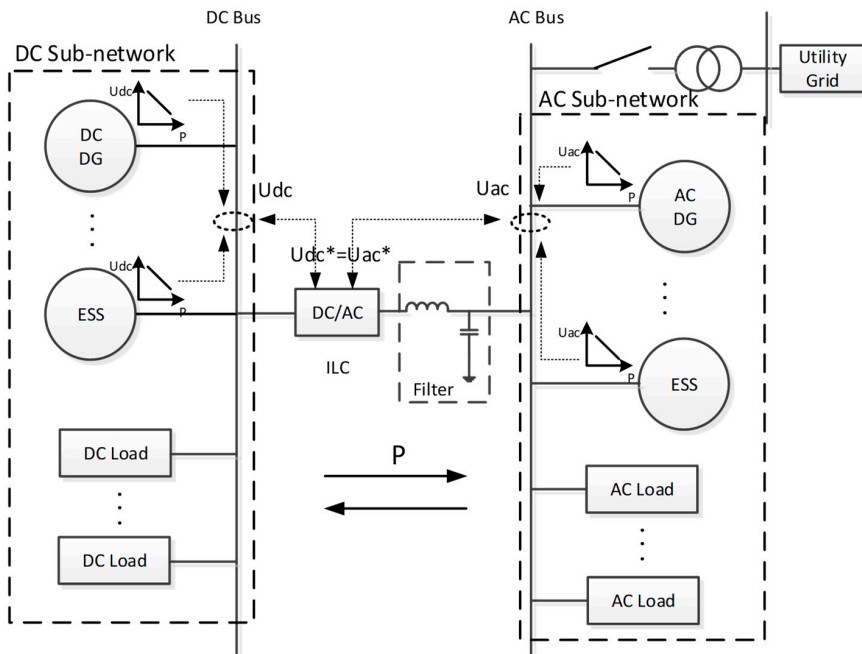

**Figure 2.** Scheme of entire network droop control.

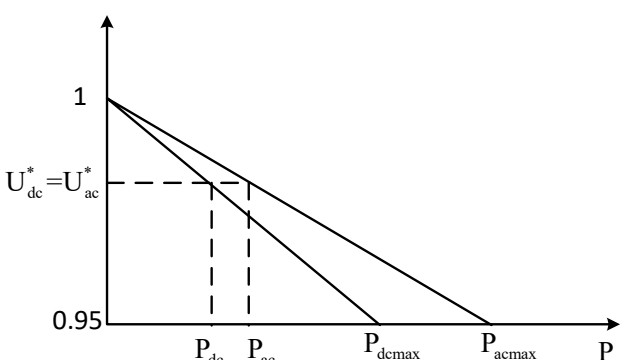

**Figure 3.** Scheme of entire network droop control.

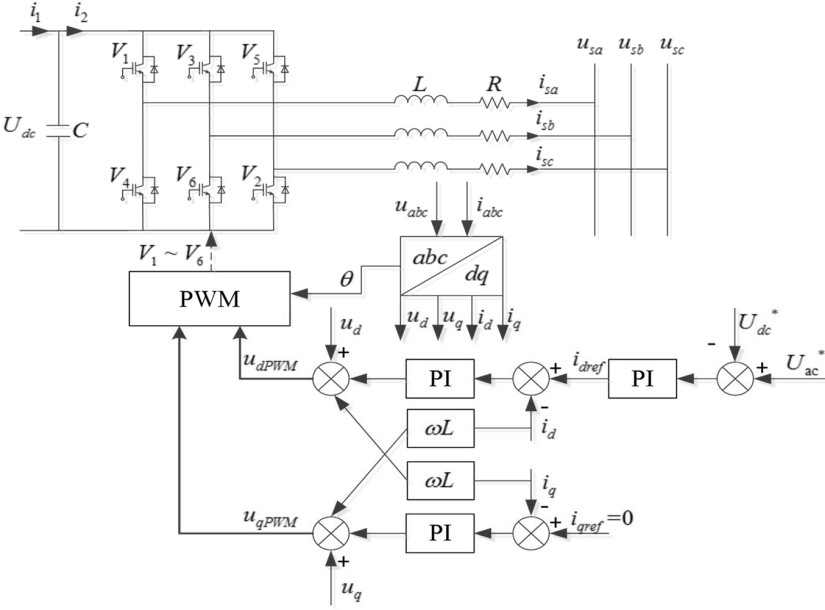

**Figure 4.** Control strategy of the ILC.

In the control loop, the *d*-axis takes AC and DC bus voltage under control, and the *q*-axis keeps the reactive power equal to 0.

## 3. A Smooth Transition Control Based on Islanding Signal

### 3.1. The Origin of Transient Impact

In the control strategy proposed in this paper, the micro-grid operates in islanding mode, while the battery and fuel cell in the DC sub-network use voltage control. The battery in the AC sub-network uses VF control. When the micro-grid operates in grid-connected mode, all of the DGs operate in power control. In order to implement different control strategies for islanding mode and grid-connected mode, each DG has two sets of controllers: a voltage controller and a power controller. The PI regulator is a very important component in both of the mode structures. The control structure of the PI regulator is shown in Figure 5, where $K_i$ is the integral gain and $K_p$ is the proportional gain.

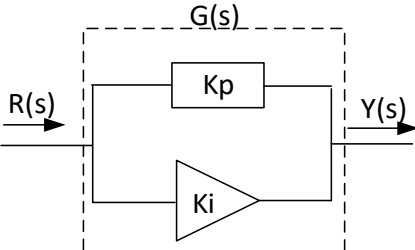

**Figure 5.** PI regulator structure.

The PI regulator's transfer function is

$$G(s) = K_p + \frac{K_i}{s} \tag{1}$$

where $G(s)$ is the gain function. $K_i$ represents the integral coefficient and $K_p$ represents the proportional coefficient.

When a micro-grid operates in islanding mode, the voltage controller is put into operation. The power controller is also working during this time, but its output control signal does not act on the converter. Similarly, when a micro-grid operates in grid-connected mode, the power controller is put into operation. The voltage controller is working at the same time, but its output control signal does not act on the converter. When the two modes are switched, due to the memory of the integral PI regulator part, there is a step change of the regulator output, which causes a sudden change in the current reference and leads to impulse current, which may cause instability. In some serious situations, it can even produce oscillation.

In addition to the impact caused by the output state of the regulator, there are voltage amplitude, phase, and frequency differences between a micro-grid and a utility grid during the switching from islanding mode to grid-connected mode, which results in a great rush current at the PCC.

### 3.2. Transient Impulse Suppression Based on Regulation State Follower

In order to solve the problem of abrupt changes of regulation state, a method based on a regulation state follower has been proposed. When the micro-grid is switched from grid-connected mode to islanding mode, DGs are switched from power control to voltage control. The output of the power control regulator should be applied as the input reference of the voltage control regulator. The output of both regulators always remains in the same state [20]. The scheme of a regulation state follower is shown in Figure 6. Here, $K_i$ (*i*=1,2,3,4) is the switch.

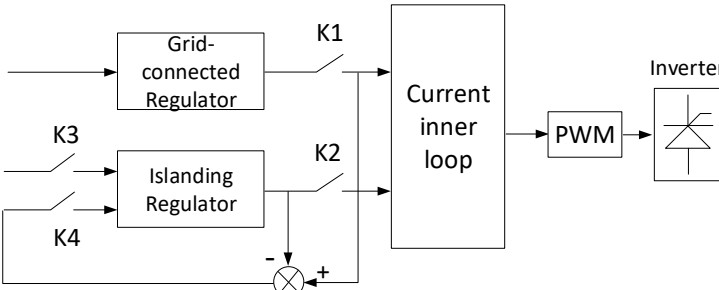

**Figure 6.** Scheme of regulation state follower from grid-connected mode to islanding mode.

During the grid-connected period, K1 and K4 are switched off, and K2 and K3 are switched on. At this time, a power controller is used to control the DG's output power. The voltage controller follows the output state of the power controller. When it is detected that the micro-grid disconnects from the utility grid, K2 and K3 are switched off, and K1 and K4 are switch on. The control mode is switched, and the micro-grid turns into islanding mode. Because the state follower maintains the output value of the voltage control regulator as equal to the output of the power control regulator, there is no abrupt change in the reference value of the inner current loop at the instant of switching, which reduces the transient impact.

When a micro-grid is switched from islanding mode to grid-connected mode, DGs need to switch from voltage control to power control. The output of the voltage control regulator is applied as the input reference of the power control regulator. In this way, the power regulator state can always follow the voltage regulator state [20]. The scheme of a state follower is shown in Figure 7.

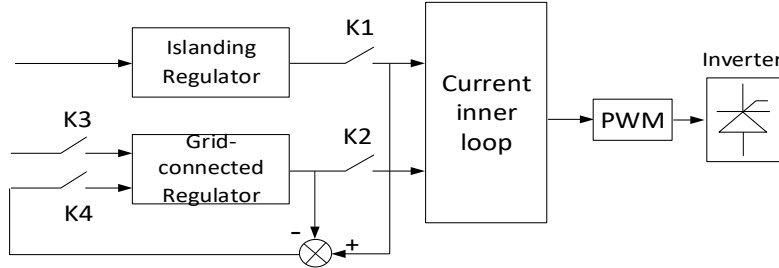

**Figure 7.** Scheme of regulation state follower from islanding mode to grid-connected mode.

The mechanism of the above two state followers is similar. During the islanding period, K1 and K4 are switched off, and K2 and K3 are switched on. At this time, the purpose of the voltage controller is to control the voltage and frequency of the AC sub-network, as well as the voltage of the DC sub-network. The power controller follows the output state of the voltage controller. When it is detected that the PCC is switched off, K2 and K3 are switched off, and K1 and K4 are switched on. The control mode is switched, and the micro-grid works in grid-connected mode. Because the state follower always keeps the output value of the power control regulator equal to the output of the voltage control regulator, there is no abrupt change in the reference value of the inner current loop at the moment of switching.

In this suppression control, the most difficult thing is to set the PI parameters. Firstly, the proportional gain $K_i$ is set to achieve a critical oscillation state. Then, the integral gain $K_p$ is calculated by formula or trial test. Now, the parameters for each controller can be settled. However, one controller has to follow the state with another, and parameter correction should be made from time to time to meet the state tracking requirement. Since the parameter correction is greatly dependent on experience, it is hard to determine an appropriate correction; even worse, such parameters do not exist in some cases, and thus the entire process takes long time.

### 3.3. Transient Impulse Suppression Based on Islanding Signal

As mentioned in the above section, the state follower-based approach suppresses the transient step by keeping an equal state value of the two controllers. The state follower can theoretically realize the synchronization of the output states of the controllers. The PI regulator is required to fulfill the outer loop control target and to implement state following. Therefore, the PI regulator settings are difficult to configure, and the control effect is not satisfied. In addition, if the operating state is frequently switched in a short period, the regulator has no time to respond, and the regulation state will change abruptly, which leads to a negative effect on the smooth transition.

In order to solve these problems, an islanding signal-based smooth transition control has been proposed in this paper. The control scheme is shown in Figure 8.

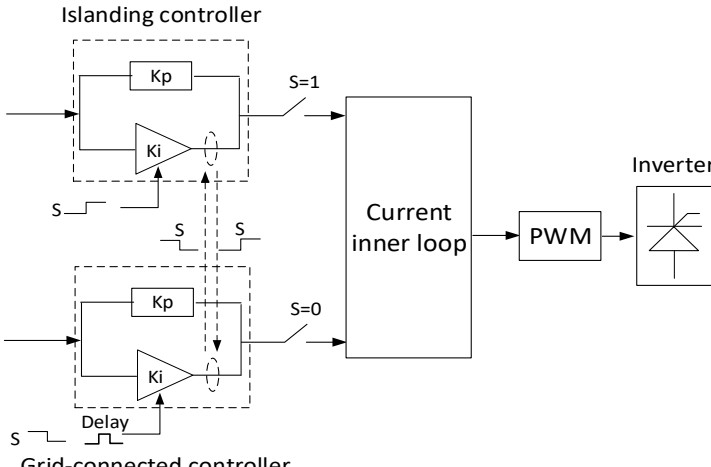

**Figure 8.** Scheme of islanding signal-based control.

The entire control consists of an outer loop and an inner loop. The output signal of the current inner loop is used to generate a control signal of the inverter. *S* represents the islanding signal. When micro-grid works in grid-connected mode, $S = 0$ and the power control regulator works. When the micro-grid works in islanding mode, *S* becomes 1, and voltage control regulator works. Figure 8 shows the outer loop of the controller. *Kp* indicates the proportional part that can respond to the input signal instantly. *Ki* indicates the integral part. The static error is eliminated by integrating the input error. The memory of the integral state is the main reason for the transient impact.

When the micro-grid switches from grid-connected mode to islanding mode, islanding signal *S* generates a falling edge from 1 to 0. This falling edge triggers the power control outer loop regulator to renew its state value. The renewed value equals to the detected state value of the voltage control outer loop regulator at that moment. When the micro-grid switches from islanding mode to grid-connected mode, the islanding signal S generates a rising edge from 0 to 1. The rising edge triggers the voltage control outer loop regulator to renew its state value. The renewed value equals the detected state value of the power control outer loop regulator at that moment. The difference is that the rising edge of the islanding signal should work on the power control regulator after a delay. Within the delay, a pre-synchronization control is performed.

In addition to the sudden change of the regulation state, a voltage and frequency difference between the micro-grid and utility grid can also cause a transient impact in the switching from islanding mode to grid-connected mode. In order to solve this problem, it is necessary to perform a pre-synchronization operation before the switch operation [20]. The pre-synchronization control scheme is shown in Figure 9.

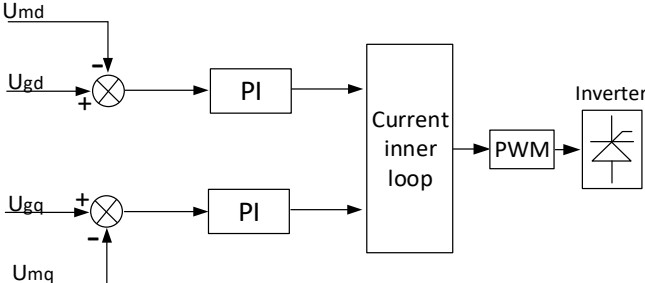

**Figure 9.** Scheme of pre-synchronization control.

Here, *Ugd* and *Ugq* refer to the voltage of the utility grid in *d-q* coordinates. *Umd* and *Umq* refer to the voltage of the micro-grid in dq0 coordinates. Through Parker transformation, the three-phase AC voltage is transformed into a DC representation in the dq0 coordinates, which avoids the direct adjustment of voltage and frequency. In addition, the transformation realizes the linear control of the voltage. When an islanding signal is detected, the pre-synchronization control is put into operation. When the voltage and frequency deviation of both sides are in the acceptable range, the PCC switches off and the pre-synchronization control stops working. The micro-grid switches to grid-connected mode successfully.

In this way, the output states of the two sets of controllers are exactly the same at the transient instant, and the impact on smooth transition can be solved. In addition, the setting of regulator parameter has been greatly simplified, since the outer loop is only employed to achieve the control target, and step three of parameter setting can be skipped.

## 4. Simulation and Discussion

### 4.1. Comparision of Differernt Transition Controls

As mentioned in Section 2, the micro-grid consists of a PV panel, a fuel cell, batteries, and AC and DC loads. The simulation model is built with PSCAD/EMTDC. The maximum load is about 300 kW. The capacity of this micro-grid is balanced in both the AC and DC sub-networks. The AC sub-network is directly connected to the utility grid through the PCC, while the DC sub-network is connected to the AC sub-network through an ILC. In this plant of a relatively large area, the distances between DGs and loads are hundreds of meters. Thus, impedance of transmission lines cannot be neglected. The line parameters are presented as AC line resistance $r = 0.5$ $\Omega$/km, inductive reactance $x = 0.35$ $\Omega$/km, and DC line resistance $r = 0.325$ $\Omega$/km. In order to sufficiently use the renewable energy, PV panels are erected on the roof of the plant. In general, taking into account the impact factors like natural shelter and area utilization, 1 kW of occupied roof area corresponds to 15 to 20 m$^2$. The plant area is about 1000 m$^2$, and the PV capacity can reach 50 kW. A fuel cell with a capacity of 50 kW serves as a controllable power supply, which is used to keep continuous supply of important loads under special conditions. The capacity of the batteries in the DC and AC sub-networks are 75 kW and 150 kW, respectively, to ensure continuous power supply to all loads for a certain period during islanding operations, and to mitigate the impact of distributed power and load fluctuations in the micro-grid. The AC bus has a rated voltage of 380 V and a frequency of 50 Hz. The AC sub-network can be directly connected to the utility grid. A unified standard for rating the DC bus voltage has not been approved yet. In the literature, the DC bus voltage is selected from 600 V to 3.5 kV [23–26]. In this paper, the rated voltage of the DC bus is selected as 800 V.

In order to validate the effect of the proposed approach, comparisons of direct transition control, traditional method state follower-based, and islanding signal-based smooth transition control are performed.

The simulation process is as follows: when $t = 0$–1.5 s, the micro-grid operates in grid-connected mode. When $t = 1.5$ s, the micro-grid turns to islanding mode, the PCC switch is disconnected, and

the control mode is changed. When $t = 3$ s, the micro-grid turns back to grid-connected mode. After pre-synchronization, the PCC is switched off and the control mode goes back to grid connection. During this process, the PV output power, DC load, and AC load are 50 kW, 112.5 kW, and 75 kW respectively.

The DC bus voltage, AC bus voltage, and frequency imply the stability of the hybrid micro-grid, which is shown in Figure 10, Figure 11, and Figure 12, respectively, by using different transition control strategies. The results of the transition controls are listed in Tables 1 and 2 for both the grid-connected to islanding case and the islanding to grid-connected case.

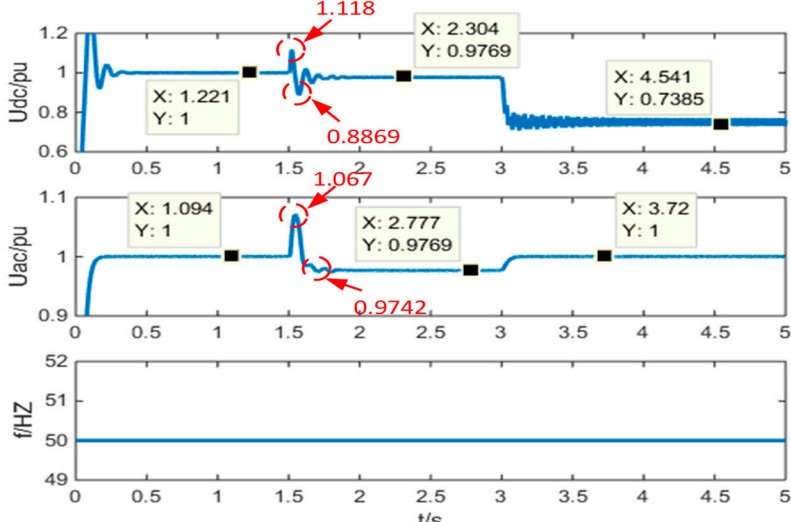

**Figure 10.** DC voltage, AC voltage, and frequency with the direct transition control.

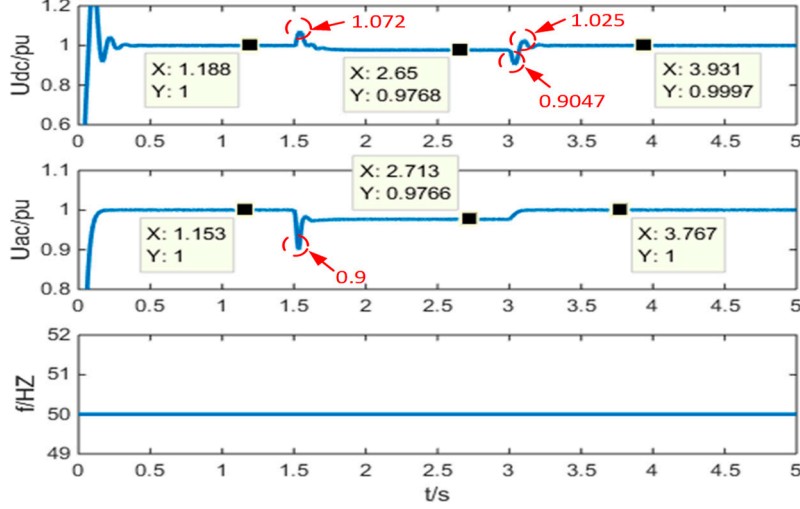

**Figure 11.** DC voltage, AC voltage, and frequency with traditional smooth transition control.

Table 1 shows the voltage offsets, switching duration, and switching result of each transition control from grid-connected mode to islanding mode. All of the three methods can obtain a successful transition, but the islanding signal-based approach performs best during the transition process. It obtains the smallest deviation of DC voltage and AC voltage, and the shortest transition time.

Table 2 illustrates a failure in transition from the islanding mode to grid-connected mode of the direct transition control. The other two approaches maintain similar results. However, the settings of the controller parameters are easily configured, as mentioned in Section 2.

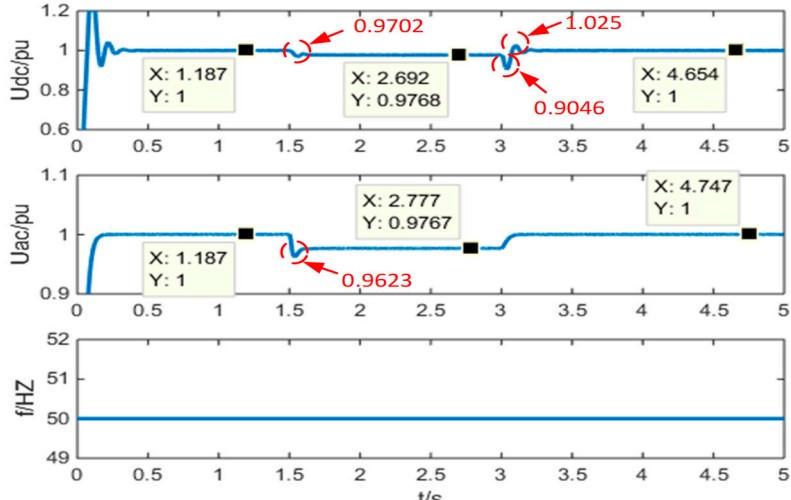

**Figure 12.** DC voltage, AC voltage, and frequency with islanding signal-based smooth transition control.

**Table 1.** Hybrid micro-grid operating status, from grid-connected mode to islanding mode.

| Transition Control | $\Delta U_{dcmax}$ | $\Delta U_{acmax}$ | Transition Time | Succeed? (Y/N) |
|---|---|---|---|---|
| Direct transition | 11.31% | 6.7% | 0.46 s | Y |
| Traditional smooth transition | 7.2% | 10% | 0.31 s | Y |
| Proposed smooth transition | 2.98% | 3.77% | 0.25 s | Y |

**Table 2.** Hybrid micro-grid operating status from islanding mode to grid-connected mode.

| Transition Control | $\Delta U_{dcmax}$ | $\Delta U_{acmax}$ | Transition Time | Succeed? (Y/N) |
|---|---|---|---|---|
| Direct transition | 26.15% | 2.31% | - | N |
| Traditional smooth transition | 9.53% | 2.34% | 0.23 s | Y |
| Proposed smooth transition | 9.54% | 2.33% | 0.23 s | Y |

In order to investigate the performance of the controller during transition, the PI regulation state for each transition approach is observed. Figures 13–15 show the regulator output state of the DGs and ILC in different transition controls.

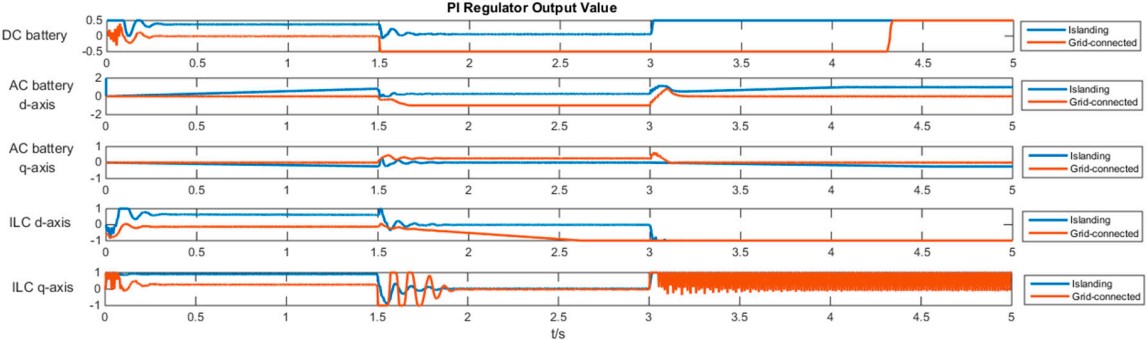

**Figure 13.** Regulation state of the direct transition control.

During $t = 0$–1.5 s, the micro-grid operates in grid-connected mode. Since the direct transition control fails to realize the regulation state following, there is a large state step at the transition instant. Therefore, the DC voltage has a large mutation, and the switching time is long. In traditional smooth transition control, a state follower is used to keep the output states of the two controllers consistent. If the grid-connected controller is working, the islanding controller will follow the output state of the grid-connected controller, as shown in Figure 14. The *q*-axis current of DC battery and ILC can provide

a successful state following. Therefore, the DC voltage offset in the traditional transition control is smaller than that in the direct transition control. However, due to the fact that the control parameters of the traditional transition controller are set for islanding operation, the regulators of the AC battery and ILC *d*-axis will become saturated, and then cannot realize state following successfully. The regulation state step changes even more than with direct transition control during the switching time. Therefore, in traditional smooth transition control, the AC voltage offset is the largest. During *t* = 1.5–3 s, the micro-grid operates in islanding mode. When the operating mode is switched at *t* = 3 s, the direct transition fails, because of the large step change in the regulator. The traditional smooth transition control achieves a good state following and realizes smooth transition, as shown in Figure 14.

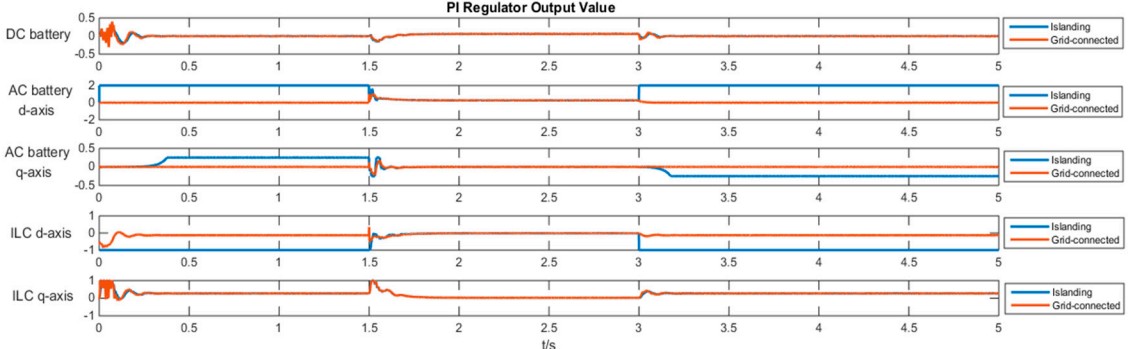

**Figure 14.** Regulation state of the traditional smooth transition control.

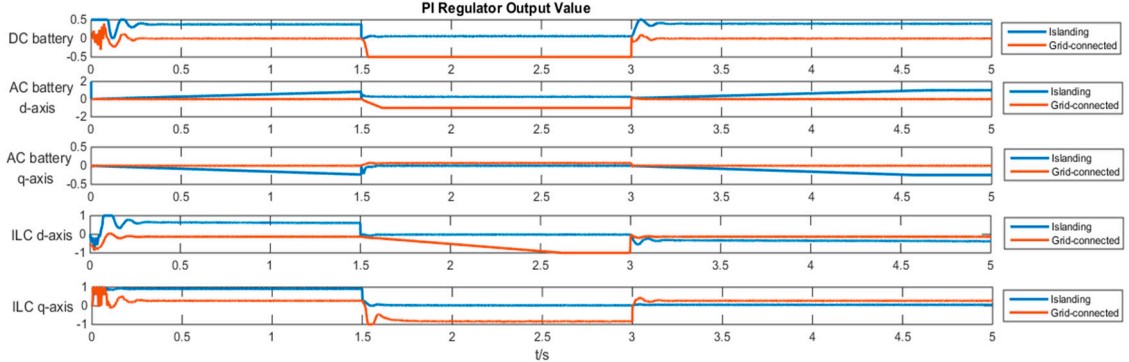

**Figure 15.** Regulation state of the proposed smooth transition control.

According to a comparison of the different approaches, it can be seen that the proposed smooth transition control can ensure non-step change of the regulation state at any transition moment to implement a smooth transition between grid-connected mode and islanding mode, as shown in Figure 15. As a result, the transient impacts during transition time is minimized regardless of the power exchange. The fluctuation is small, and the time to reach a new stable state is short.

### 4.2. Influence of Load Change on Smooth Transition

The AC/DC hybrid micro-grid is more complex than the traditional micro-grid, and the energy can flow freely through the ILC in the AC and DC sub-networks. Based on the proposed smooth transition control, this section discusses the impact of load variation on smooth transition.

The simulation process is as follows: when *t* = 0 s, the micro-grid starts in grid-connected mode. When *t* = 1.5 s, the PCC switches on, and the micro-grid turns to islanding mode. When *t* = 3 s, the PCC switches off, and micro-grid turns to grid-connected mode again. During the process, the PV output power is 50 kW, the DC load is 112.5 kW, and AC loads are 30 kW, 75 kW, 120 kW, and 150 kW, respectively. The waveforms of DC voltage, AC voltage, and ILC transmission power are shown in Figures 16–19.

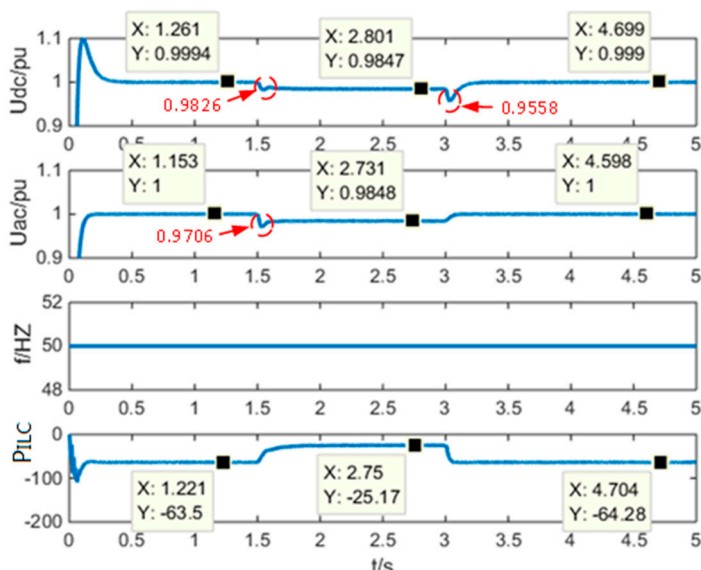

**Figure 16.** DC voltage, AC voltage, frequency, and ILC transmission power (DC load = 112.5 kW; AC load = 30 kW).

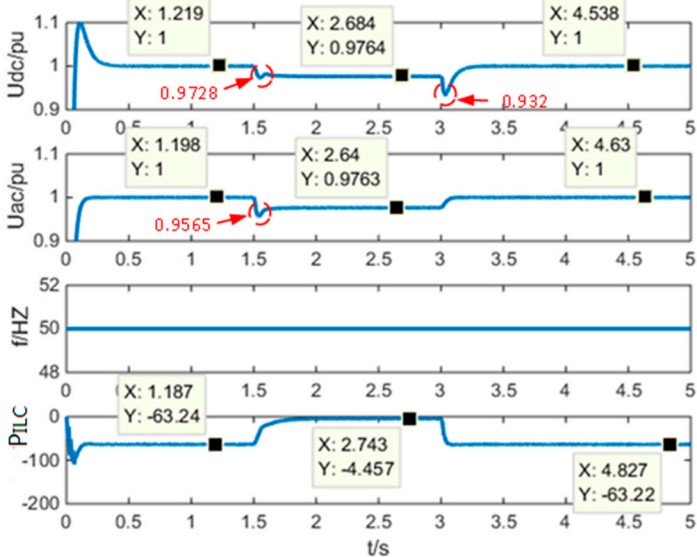

**Figure 17.** DC voltage, AC voltage, frequency, and ILC transmission power (DC load = 112.5 kW; AC load = 75 kW).

In Table 3, the DC load, AC load, PV output power, ILC transfer power at the instant of mode transition, DC voltage maximum change from grid-connected mode to islanding mode, DC voltage maximum change from islanding mode to grid-connected mode, and maximum AC voltage variation are listed.

**Table 3.** AC/DC hybrid micro-grid operating state.

| DC Load/kW | AC Load/kW | $P_{pv}$/kW | $\Delta ILC$/kW | $\Delta Udc1$ | $\Delta Udc2$ | $\Delta Uac$ |
|---|---|---|---|---|---|---|
| 112.5 | 30 | 50 | 38.33 | 0.24% | 2.96% | 1.4% |
| 112.5 | 75 | 50 | 59.04 | 0.35% | 4.53% | 2.04% |
| 112.5 | 120 | 50 | 79.85 | 0.42% | 6.61% | 2.37% |
| 112.5 | 150 | 50 | 93.48 | 0.47% | 7.69% | 2.48% |

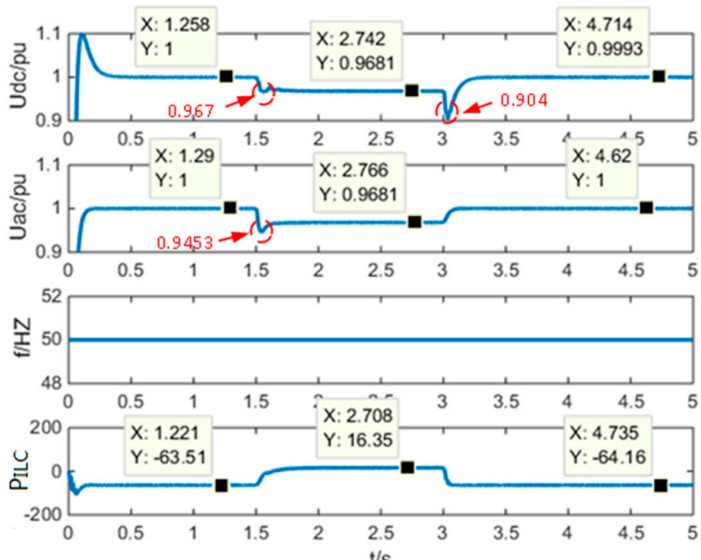

**Figure 18.** DC voltage, AC voltage, frequency, and ILC transmission power (DC load = 112.5 kW; AC load = 120 kW).

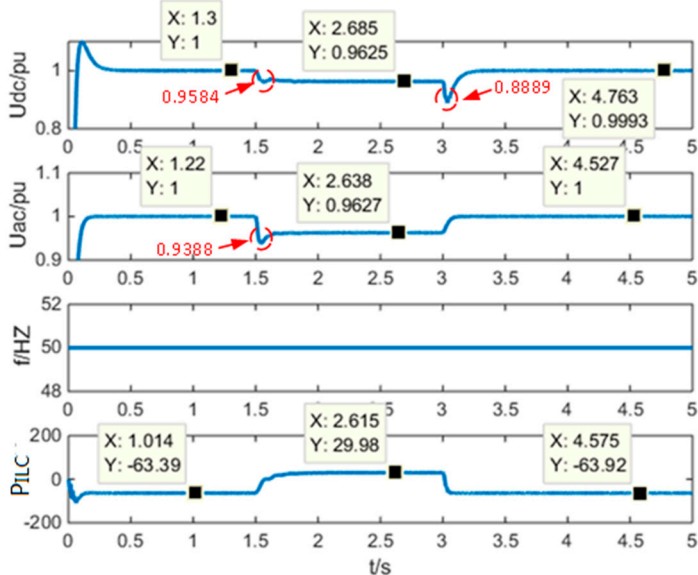

**Figure 19.** DC voltage, AC voltage, frequency, and ILC transmission power (DC load = 112.5 kW, AC load = 150 kW).

Since PV output power and DC load do not change during the process, the transmission power of ILC does not change during grid-connected operation, and equals the power absorbed by the DC sub-network. With the increase of the AC load, the transmission power of ILC increases during islanding operation. Thus, the power changes during the transition instant, resulting in an increase of voltage offset during transition process, as shown in Figure 20. In summary, load variation will affect the smooth transition of hybrid micro-grid. The greater the unbalanced power in the AC/DC sub-network, the greater the negative impact on smooth transition takes.

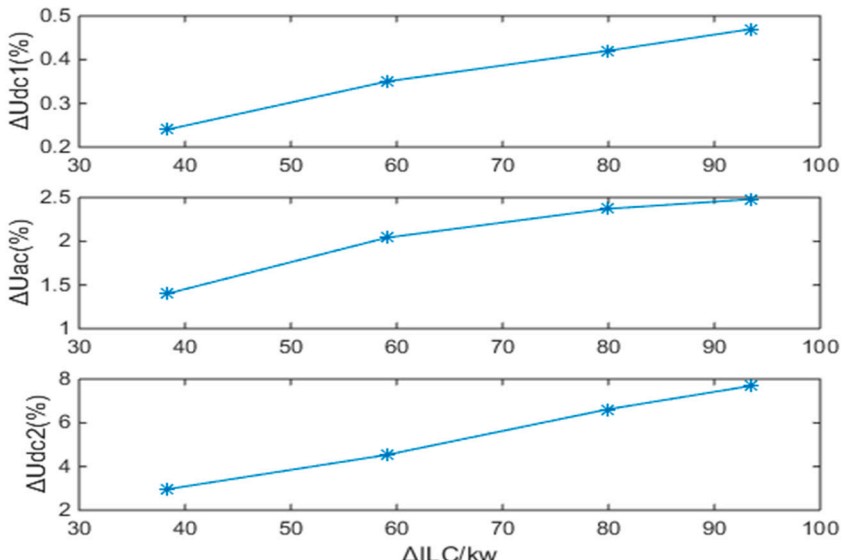

**Figure 20.** Maximum variation of AC and DC voltage.

## 5. Conclusions

In this paper, an islanding signal-based smooth transition in a low-voltage, hybrid micro-grid has been discussed. The state of the islanding signal is detected for transition control, which facilitates the settings of the controller, and the step change of the regulator is greatly suppressed to obtain a smooth transition. The parameter update of controllers is triggered by the detection of an islanding signal, and remains consistent during the transition. A smooth transition is implemented in the condition of various power exchanges between the micro-grid and utility grid. In contrast to the direct transition approach and traditional smooth transition control, the proposed approach shows the least voltage deviation and the shortest switching process during transition.

**Author Contributions:** Conceptualization, Z.C. and T.Z.; Methodology, Z.C. and T.Z.; Software, Z.C., T.Z. and C.L.; Validation, Z.C., T.Z. and C.L.; Formal Analysis, Z.C., T.Z. and C.L.; Investigation, Z.C., T.Z. and C.L.; Resources, Z.C., T.Z. and C.L.; Data Curation, Z.C., T.Z. and C.L.; Writing—Original Draft Preparation, Z.C., T.Z. and C.L.; Writing—Review & Editing, Z.C., T.Z. and C.L.; Visualization, Z.C., T.Z. and C.L.; Supervision, Z.C., T.Z. and C.L.; Project Administration, T.Z.

**Funding:** This research was funded by the key Research and Development program of Shaanxi Province. grant number 2017ZDCXL-GY-02-03

**Conflicts of Interest:** The authors declare no conflict of interest.

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
