# Peer review of "An Islanding Signal-Based Smooth Transition Control in AC/DC Hybrid Micro-Grids"

_applsci, doi:10.3390/app9142804_

Round 1
Reviewer 1 Report
The paper aims at developing a methodology to ensure a bumpless transition of an AC/DC microgrid from the on-grid to the off-grid and “vice versa”. Even if the reviewer is very interested in this topic, some major issues needed to be addressed before the publication.
Comments and recommendations are as follows:
The Introduction should be improved. The current state-of-art must be clearly presented in order to better point out the contribution of this paper. I would suggest to break the introduction to three sub-sections: Motivation, literature review, and contributions.
In the literature review, firstly a reference to the already existing pilot projects for smart microgrids should be presented. In particular, a clear definition of the main possible states and transitions in which a microgrid can be operated should be provided. In this sense useful could be the following paper: “Prince—Electrical Energy Systems Lab: A pilot project for smart microgrids”.
Then, the main issues arising from the sudden loss of the main grid support as well as the main methods developed in technical literature to overcome them should be clearly discussed. Useful could be the paper “A simple circuit model for the islanding transition of microgrids”.
For what concerns the technical part there are some issues that needed to be addressed.
a. What is the energy storage capacity of the fuel cell?
b. What is the energy storage capacity of the battery?
The reviewer has understand that the microgrid under consideration is based on droop-control. For this reason no isochronous controller is devoted within the hybrid microgrid to provide the voltage and frequency reference. Moreover, when the hybrid microgrid operates in isolated mode, none of the two subnetworks operates as the master for the other one. In fact, they operate separately among them and through their own droop-controlled converters, they keep their voltage and frequency at their nominal values. Anyway, in doing this there may be a slightly difference in voltage and frequency at both sides of the ILC and thus it can be seriously damaged. Please, could you better explain how to overcome this issue?
On page 4, line 131 Kp should be the proportional gain and not the integral gain. In addition, Ki should be the integral gain and not the proportional one.
Conclusions should be improved. In particular, the results obtained from the tests carried out on the microgrid should be discussed in order to carried out the benefits deriving from the adoption of the proposed method.
Author Response
Thank you for your helpful comments. We have carefully revised the paper, especially on the points you have addressed.
1. The introduction has been improved following the steps of motivation, literature review, and contributions. In the literature review, references to the already existing pilot projects for smart microgrids have been presented. The state-of-the-art of transition schemes have been addressed with additional references as presented in line 55-71. The contribution of this paper has been emphasized in line 79-88. In particular, two references of “Prince—Electrical Energy Systems Lab: A pilot project for smart microgrids” and “A simple circuit model for the islanding transition of microgrids” have been referred to in this revised version.
2. The storage capacity of fuel cell and battery in the simulation is denoted in line 248 and line 250. The storage capacity will affect the output power and system voltage. There will be a slight voltage drop due to discharge. This paper focuses on the transition behavior of a hybrid micro-grid. Usually, the transition happens instantly. During such a short period, the output voltage of fuel cell and battery can be considered unchanged. Herein, we assume that in the transition time, the fuel cell and battery work under the rated condition, and the storage capacity is with infinite amount.
3. Hybrid micro-grid consists of AC subnetwork and DC subnetwork. These two subnetworks are connected with ILC and the power management is realized through an entire network droop control. In the DC subnetwork, only voltage matters, so there is no frequency difference between both sides of ILC. For the whole system involving hybrid micro-grid and utility grid, voltage and frequency difference problem do exit. To solve this problem, pre-synchronization control has been applied, as discussed in line 217-235.
4. We have revised the following notation of parameters: Kp is the proportional gain and Ki is the integral gain.
5. Conclusion has been revised to clarify the advantages of this approach (section 5).
Reviewer 2 Report
The paper is interesting and clear in the topic. The main weakness is that it focuses on theoretical applications and does not provide effectiveness in practical applications.
In the introduction section, it is suggested to cite a large number of papers on hybrid AC/DC microgrids, by highlighting the need of improvements in this field.
Provide more details on the contributions of this paper compared to the current literature.
Provide more discussion on the actual application of hybrid AC/DC microgrids and, particular, the need of the proposed control.
Line 111: is “unitizing” correct?
Line 165: describe the adopted state synchronization.
How did you simulate the Transition Controls? Simulink? Please discuss this aspect and provide the information needed to replicate the simulations.
All the information on data and parameters on the case study should be included in the simulation section, instead that in the theoretical part of the paper (i.e., section 2)
Improve conclusion section, by better detailing the advantages one can obtain by the proposed approach.
Author Response
Thank you for your pertinent comments. We have thoroughly revised the manuscript following your suggestion.
1. In introduction, we have discussed the state-of-the-art of AC/DC micro-grid transition under the investigation of numerous references in this fields. Comparison has been made to show the necessity and improvement of smooth transition control in this paper. The revised context is highlighted in yellow color in line 55-71 and line 79-88.
2. All the information for simulation and related parameters of the hybrid micro-grid have been described in detail in the revised version, as presented in section 4.1(line 238-256).
3. The word “unitizing” was not correct and we have revised the corresponding statement in line 115 in a new way. And “state synchronization” made some misunderstanding and we have clarified the presentation in line 169-170 with new statements.
4. Conclusion has been carefully revised (section 5).
Round 2
Reviewer 1 Report
The revised paper has been significantly improved
Reviewer 2 Report
In the new version of the paper, the Authors provided some required improvements.